# Combined Femtosecond Laser Glass Microprocessing for Liver-on-Chip Device Fabrication

**DOI:** 10.3390/ma16062174

**Published:** 2023-03-08

**Authors:** Agnė Butkutė, Tomas Jurkšas, Tomas Baravykas, Bettina Leber, Greta Merkininkaitė, Rugilė Žilėnaitė, Deividas Čereška, Aiste Gulla, Mindaugas Kvietkauskas, Kristina Marcinkevičiūtė, Peter Schemmer, Kęstutis Strupas

**Affiliations:** 1Femtika Ltd., Keramikų Str. 2, LT-10233 Vilnius, Lithuania; 2Laser Research Center, Vilnius University, Saulėtekio Ave. 10, LT-10223 Vilnius, Lithuania; 3General, Visceral and Transplant Surgery, Department of Surgery, Medical University of Graz, Auenbruggerplatz 29, AT-8036 Graz, Austria; 4Faculty of Chemistry and Geosciences, Vilnius University, Naugarduko Str. 24, LT-03225 Vilnius, Lithuania; 5Institute of Clinical Medicine, Faculty of Medicine, Center of Visceral Medicine and Translational Research, Vilnius University, M. K. Čiurlionio g. 21, LT-03101 Vilnius, Lithuania

**Keywords:** selective laser etching, 3D laser microfabrication, laser welding, glass microfluidics, femtosecond laser microprocessing

## Abstract

Nowadays, lab-on-chip (LOC) devices are attracting more and more attention since they show vast prospects for various biomedical applications. Usually, an LOC is a small device that serves a single laboratory function. LOCs show massive potential for organ-on-chip (OOC) device manufacturing since they could allow for research on the avoidance of various diseases or the avoidance of drug testing on animals or humans. However, this technology is still under development. The dominant technique for the fabrication of such devices is molding, which is very attractive and efficient for mass production, but has many drawbacks for prototyping. This article suggests a femtosecond laser microprocessing technique for the prototyping of an OOC-type device—a liver-on-chip. We demonstrate the production of liver-on-chip devices out of glass by using femtosecond laser-based selective laser etching (SLE) and laser welding techniques. The fabricated device was tested with HepG2(GS) liver cancer cells. During the test, HepG2(GS) cells proliferated in the chip, thus showing the potential of the suggested technique for further OOC development.

## 1. Introduction

Nowadays, a remarkable idea in medical treatment and diagnostics is that of lab-on-chip (LOC) devices, which allow the miniaturization of massive diagnostic tools and the reduction of testing on live organisms. The LOC technology has become more and more promising as an innovation in the research field because of such devices’ functionality. Usually, a single chip is integrated with one or several laboratory functions, allowing laboratory processes to proceed quickly with high precision. In combination with the working principles of microfluidics, these devices show potential in health applications, medicine, and more. The designs and applications go in various directions—for example, injection-molded polymeric LOC for blood plasma separation [1], plastic LOC for herbicide residue monitoring in soil [2], and devices for the detection of various pathogens in food [3,4], for the detection of viruses [5,6], and for the detection of bacteria [7]. The applications of LOC devices do not end here; they could be used for research on various cells, such as research on the growth of fungal cultures [8] or on cancer and tumor cells [9,10,11]. LOC devices could also be used for tissue or organ research. These devices are usually called organ-on-chips [12] or organoids-on-chips (OOCs); as the name suggests, these are chips with an organ/organ tissue that is grown inside the chip and will later be used for various tests, such as tests on drugs [13,14], toxins [15], and others [16]. One example is a liver-on-chip device [17]. OOC devices are becoming more popular because of their versatility and advantages in comparison with usual testing methods. LOC devices are rapidly merging directions. However, this is still an open area for innovations—from new materials to innovative design research.

The dominant technology in the production of such devices is molding [18,19]. On one hand, molding is a very effective and cheap technique for mass production [20]. Molded chips can cost a few euros per piece. On the other hand, the price can increase by many times (even hundreds of times) for prototyping when new master structures are needed for any changes in the chip design. We propose an alternative solution based on ultrafast laser material processing in this work. Even though laser micromachining is quite an expensive process, with which the price of a chip could be in the range of tens to hundreds of euros, it is a good tool for the prototyping of complex devices. Femtosecond-pulse lasers are a powerful tool and bring a few significant advantages for material processing, such as high precision and high quality. In addition, this tool leads to entirely new microprocessing techniques based on nonlinear material–light interactions, which are impossible with other tools. One example is the multi-photon polymerization [21,22] technique, which enables the fabrication of hundreds of 3D structures with nanometer precision and resolution out of polymers. Furthermore, a femtosecond laser is an excellent tool for glass microprocessing. Due to nonlinear light–material interactions, glass can be modified directly in the volume without damaging its surface [23]. A few different modifications can be created inside a volume of glass: changes in the refractive index [24], nanogratings [25], or microvoids [26]. The type of modification induced depends on the radiation intensity used [24]. However, each type of modification can be used for the microprocessing of different materials. For instance, by inscribing a refractive index or nanogratings, refractive optical elements can be formed [27]. A combination of the inscription of nanogratings with subsequent selective laser etching (SLE) could be implemented [28,29]. Meanwhile, microvoid modifications were formed during laser ablation [30] to remove a material directly or during the laser welding process [31] to bond two materials together in their contact. Therefore, many different tasks could be accomplished with a single femtosecond laser source.

This study demonstrates a combination of a few different femtosecond glass microprocessing techniques for liver-on-chip device manufacturing. Here, we combine the selective laser etching and welding techniques to produce a liver-on-chip device. At the end, we provide the results of tests on the manufactured devices.

## 2. Materials and Methods

The main idea was to create an OOC prototype that is suitable for liver-on-chip testing by using femtosecond laser microprocessing methods. First of all, the concept of the device design was chosen. The central principle was to create a microfluidic system with three individual channels that were separated in the center with filters. The center channel needed to be filled with liver cells. Meanwhile, the side channels could be filled with other materials, such as drugs or different types of cells. The reactions between liver cells and the drug could be observed in the mentioned chip area. A similar design concept with pillar filters has already been published elsewhere [32]. However, the design of the microfluidic channels was significantly changed. The critical requirements in microfluidics are keeping channels as short as possible and avoiding microwells, dead-ends, or sharp forms in the channels. The microfluidic chip design that was created is shown in Figure 1. These adjustments make microfluidic systems more user friendly because the channels can be filled without sophisticated microfluidic pumps. In the described experiments, simple lab pipettes were used to manipulate substrates and fill the designed chip.

For liver-on-chip fabrication, two different laser microprocessing methods were used. Plates with channel systems and integrated filters were fabricated with the SLE method. SLE is a technology that enables the production of 3D structures out of solid-state transparent materials [33,34]. The implementation of SLE consisted of several steps. First, laser-induced periodic modifications called nanogratings were formed in the volume of the material by using ultrashort pulses. Subsequently, a laser-modified material was etched out with aggressive etchants, such as hydrofluoric acid (HF) or potassium hydroxide (KOH) [35]. SLE was performed on amorphous UV-grade fused silica (UVFS) with a 1 mm thickness. The UVFS substrates were chosen for the fabrication of the channel system. Laser microfabrication was performed by using a Laser NanoFactory workstation (Femtika Ltd., Lithuania). The utilized workstation was equipped with a Yb:KGV femtosecond laser (Pharos, Light Conversion Ltd., Lithuania). For the SLE experiments, a fundamental wavelength of 1030 nm, a 700 fs pulse duration, and a frequency of 610 kHz laser radiation was used. The laser radiation was focused with 20 × 0.45 NA Nikon focusing objective equipped with automated aberration correction (add-on device from Femtika Ltd., Vilnius, Lithuania). Within specific radiation exposition conditions, modifications in porous materials called nanogratings could be inscribed inside the volume of the glass [25]. Subsequently, after inscribing particular material modifications, the sample was etched in a potassium hydroxide (KOH, Eurochemicals, Lithuania) solution at a 6 mol/L concentration with distilled water at a temperature of 90 °C. The etching protocols were optimized for the fastest etching procedure. This optimization has already been published elsewhere [36].

Since we needed an encapsulated microfluidic channel system, the channels formed on the plates had to be sealed. Another femtosecond-laser-radiation-based technique—laser welding [37,38]—was used for that. Only the contact between two plates could be affected without damaging the surface by using high-intensity radiation due to nonlinear light–material interactions. With high power, the material in the contact could be melted, and a firm connection was formed in the mentioned samples. The laser welding part was carried out with the same Laser NanoFactory workstation. The laser welding experiments are conducted with the same 1030 nm wavelength, a pulse duration of approximately 200 fs, and a 610 kHz pulse repetition rate. The laser radiation was focused with a 0.5 NA aspherical lens. After the etching and before the welding, the samples were washed in a piranha solution (4:1 *v/v* of sulfuric acid (95–98%, Sigma-Aldrich, Darmstadt, Germany) and hydrogen peroxide (50%, Sigma-Aldrich, Darmstadt, Germany), respectively). Afterward, the samples were rinsed in distilled water and isopropanol. The basic scheme of the fabrication of the chips is shown in Figure 2.

The manufactured liver-on-chip prototypes were rinsed with sterilized aqua dest, submerged in 70% ethanol solvent for initial disinfection, and UV irradiated for 20 min for further sterilization. The chips were coated with poly-L-lysine polymers to promote cell adhesion. The coating was performed by filling the chamber with 10 µL of poly-L-lysine solution (Sigma Aldrich, Vienna, Austria) through a seeding channel, followed by an incubation period at 37 °C until they had completely dried. These prepared liver-on-chip systems were stored in standard sterile conditions until use.

The human liver HCC cell line HepG2(GS) (originating from ATCC, Rockville, MD, USA) was used for biocompatibility testing. These cells were cultivated under standard conditions (37 °C, 5% CO${}_{2}$ in a humidified atmosphere) in MEM media (Gibco, Thermo Fisher Scientific, Vienna, Austria) supplemented with 10% fetal bovine serum (GE Healthcare Life Sciences, UT, USA) and 1% penicillin/streptomycin (Sigma Aldrich, Vienna, Austria). The media were renewed every other day, and cells were passaged once they grew to 80–90% confluence. The chambers of the liver-on-a-chip systems were filled with 10 µL of a cell suspension to reach 2 × 10${}^{4}$ HepG2(GS) cells/cm² through the seeding channels. The chips were submerged in culture media in a Petri dish and placed in a cell incubator under standard conditions (37 °C, 5% CO${}_{2}$ in the humidified atmosphere) overnight to avoid evaporation. Then, cell adherence and growth were determined through daily microscopic examination. The media were renewed every other day in a standard manner. Finally, the Trypan Blue exclusion test was conducted to further confirm cell viability. The total testing time frame for a single sample was nine days. This experiment was repeated three times in separate pieces. Every sample was used for a single experiment and then disposed of as waste due to the biological exhaustion of the sample.

## 3. Results and Discussion

### 3.1. Fabrication of the Microfluidic Chips

We began the research by producing microfluidic chips with integrated filters on a glass substrate. A picture of a fabricated chip and SEM pictures of its particular parts are shown in Figure 3. In general, the filter was a row of elliptical glass pillars. The dimensions of every pillar were a width of 36 µm and a length of 55 µm. The spacing between each pillar was 14 µm, corresponding to the filter’s pores. The height of the fabricated pillars was 200 µm, which was identical to the depth of the microfluidic channel. The filters were integrated into a channel that was 5 mm long and 0.9 mm wide. In the mentioned area, two rows of pillars divided that zone into three distinguished channels; the side channels were 200 µm wide, and the central one was 400 µm wide.

Visually, smooth features and a high channel aspect ratio were obtained. However, not only modified but also unmodified material was etched during the etching process. Thus, this led to widened channels and features. The mentioned effect created a critical limitation of the aspect ratio, which is especially important in microfilter fabrication. Since the cells were micro-scale objects, the precision of the integrated filter was essential. In the model, the gap between pillars shown in Figure 3c,g was a single-line inscription, which was a minimal laser modification between pillars. However, after the etching, the gap between pillars tended to increase up to 14 µm, which limited the accuracy and minimal features of the filter. In LOC applications, the surface quality of the channel is also critical. This feature affects liquid flow. The higher roughness of the surface, the more friction it creates. Thus, the channels become difficult to fill due to the higher surface roughness. The surface quality of the processed channels was evaluated with an optical profilometer. The surface roughness of the etched channels was around 250 nm root mean square (RMS). The surface topology of the channel is shown in Figure 3b.

### 3.2. Welding of the Microfluidic Chips

The chips were sealed through laser welding. Optical contact was required to create firm contact between plates. That meant that the gap between two surfaces should be a few times smaller than the wavelength used, which was 1030 nm. To achieve that, the high cleanness and surface quality of the samples was needed. Thus, the samples were washed in piranha solution to remove all organic remains from the chip. Before welding, the chip was rinsed in isopropanol and distilled water. Two glass plates were put on each other when the chip was still wet. Isopropanol is a liquid that forms a low contact angle with glass [36], which means that it wets surfaces. Thus, due to the wet contact with isopropanol, the two plates tended to have a smaller gap between each other, which enhanced the welding quality. The two plates were welded in contact everywhere around the channel system without damaging or affecting the channels or the filers themselves. Welding seams were made within a spacing of 100 µm. An image of the welding seams around the microfluidic channels is provided in Figure 4b. An optical picture of the produced chip is shown in Figure 4a.

It was already demonstrated that the welding strength could be close to the mechanical strength of the bulk material [38]. However, we performed an additional experiment to test the welding quality. The main idea was to affect the chip with a particular force and test what forces led to the breaking of the chip. This experiment is shown in Figure 5a,d. The force was gradually increased, and the chip broke when it was affected by a 9.3 N force. A broken chip after the test is presented in Figure 5b,e. After the sample was broken, the chip was destroyed. Nonetheless, in most places, the welding seam still held the two glass plates together. Pictures were taken of the welding seam from the side of the broken chip Figure 5c,f. The welding seam seemed to have small periodic cracks; however, the material in between seemed to be completely fused between the two plates. The provided material contributed to the statement that the strength of the welding seam was comparable to the strength of the bulk material. The welding seam was not the weakest part of the chip. The chip tended to break through the channels, which were the most fragile parts of the chip.

### 3.3. Testing of the Liver-on-Chip’s Functionality

The prototype device’s functionality was tested with liver cells. A homogeneous film of poly-L-lysine was formed on the surface of the liver-on-chip systems. Uncoated chips were also tested; however, HepG2(GS) did not adhere due to the smooth chips’ surface, while cells attached well on the coated surfaces. A total of 24 h after seeding, HepG2(GS) cells started to grow in a monolayer. The number of spheroids and size were consistent with the starting cell density. During the media change, the cell monolayers remained adherent to the surface and continuously grew. After 96 h of culture, the HepG2(GS) cells created large irregular spheroids. After 7 days of culture, the growth of spheroids led to the formation of large clusters of spheroids in the whole liver-on-ship system. A picture of the mentioned test is shown in Figure 6. From day 7, the cell viability decreased, as seen from the roundish shape of some of the cells. After 9 days, most of the cells had died according to the Trypan Blue test; this can be seen in Figure 6i,j. In general, the Trypan Blue exclusion test showed that the adhesive cells had good viability for 7 days of the experimental period.

### 3.4. Future Prospects

The results show that femtosecond laser glass microfabrication is suitable for the development of liver-on-chip devices. However, there is still space for manufacturing improvements and more flexible design implementations. One of the observed challenges was the filter’s accuracy and the minimum pore size. For example, the accuracy of SLE-made features depends on the etching rate between the modified and unmodified material. With the currently used chip design and etching protocol, the minimum spacing between pillars is 14 μm. Therefore, during the tests, it was noticed that a fraction of cells could pass through the filter and appear in other channels. Thus, the mentioned spacing between the pillars must be reduced to keep all of the cells inside the center channels. Therefore, the accuracy and sharpness of the structural etching rate of the modified material can be increased. This can be achieved in a few ways, such as by optimizing the etching process, the etching solution [36], and the laser parameters [39], or by introducing a specific femtosecond burst regime [40]. Such improvements can create the possibility of tuning the filter size more accurately. Moreover, the surface quality of the fabricated chips can be increased by applying additional post-processing, such as heat treatment [41,42] or CO${}_{2}$ laser annealing [43]. Such enhancements should decrease the friction between the surface of the channels and substances inserted in the chip, making it easier to fill the chip and test its functionality.

## 4. Conclusions

Here, we demonstrated a flexible method for microfluidic prototyping. A microprocessing tool with a single femtosecond laser source was used to produce microfluidic channel systems with integrated glass pillar filters. The channels were hermetically sealed by welding a glass plate on top of the channels with the same femtosecond processing tool. In this way, the whole liver-on-chip device was manufactured with a single workstation. Afterward, the produced chips were tested as liver-on-chip devices by filling the central channel with HepG2(GS) liver cancer cells. The cells tended not to adhere to uncoated glass surfaces. However, the cells aggregated on channels coated with a homogeneous film of poly-L-lysine, making it possible to test other cells’ reactions to various stimuli introduced to the chip. These experiments show that femtosecond glass microprocessing is a potential and attractive technique for developing liver-on-chip devices. Here, we showed a potential platform for HepG2(GS) liver cancer cell testing; however, some improvements to these technologies still need to be made.

## Figures and Tables

**Figure 1 materials-16-02174-f001:**
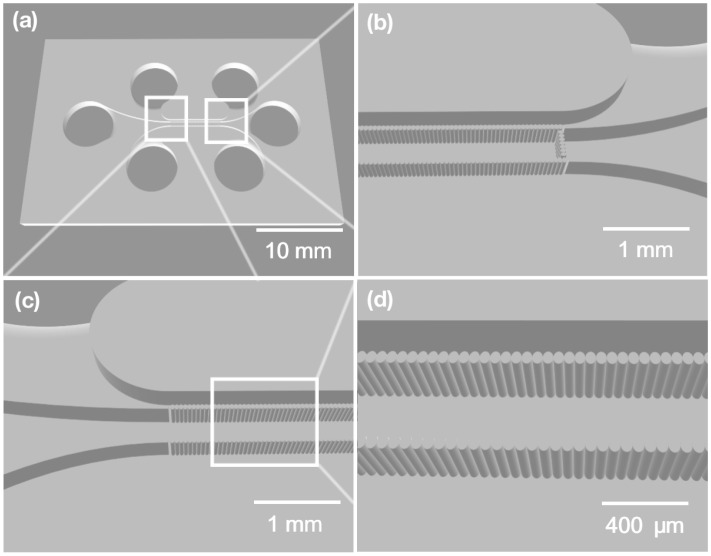
A picture of the liver-on-chip model that was created. (**a**) A full view of the chip model; three cylindrical holes serve as inlets for each channel, and three cylindrical holes are outlets for each of the three channels. All three channels meet in the center and are separated by a pillar-type filter. (**b**,**c**) An enlarged view of specific parts of the channel system. (**d**) Glass-type filters in the center of the chip.

**Figure 2 materials-16-02174-f002:**
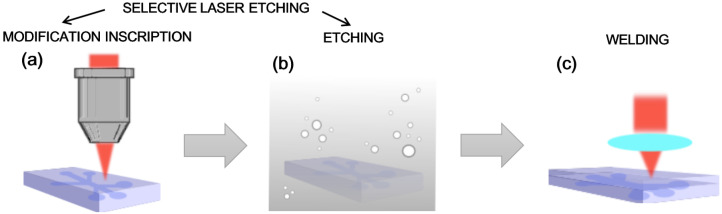
A basic scheme of the laser processing of the lab-on-chip devices. (**a**) Inscription of nanogratings on the glass plate by focusing light with a microscopic objective. (**b**) Subsequent etching of the laser-processed sample. (**a**,**b**) Presentation of the SLE technique. (**c**) The laser welding process used to seal the channel system with an additional glass plate.

**Figure 3 materials-16-02174-f003:**
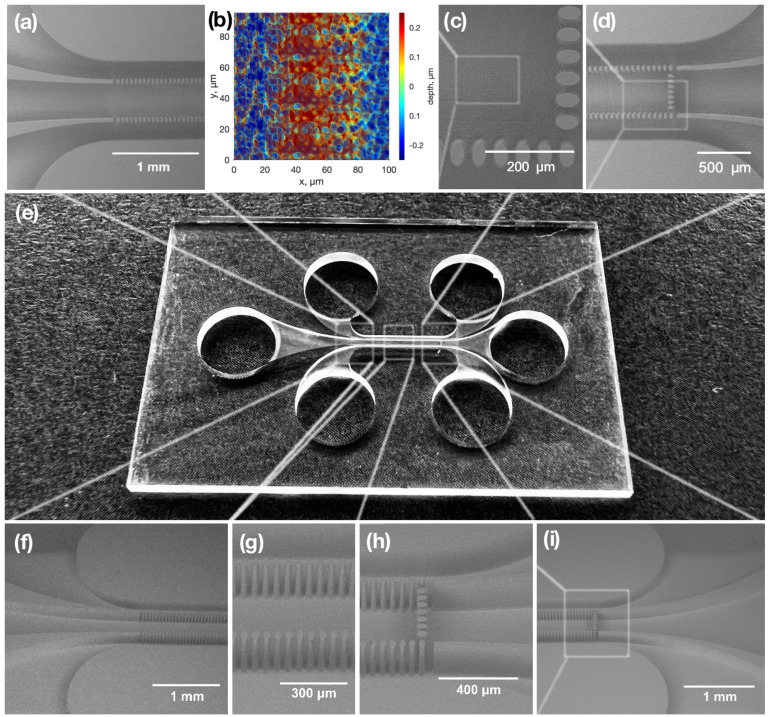
Pictures of the produced liver-on-chip device. (**e**) Optical picture of a full device. (**a**,**c**,**d**) SEM images of the tops of the specified chip parts. (**f**–**i**) SEM images of specified chip parts at 45°. (**b**) Surface topology of the channel surface.

**Figure 4 materials-16-02174-f004:**
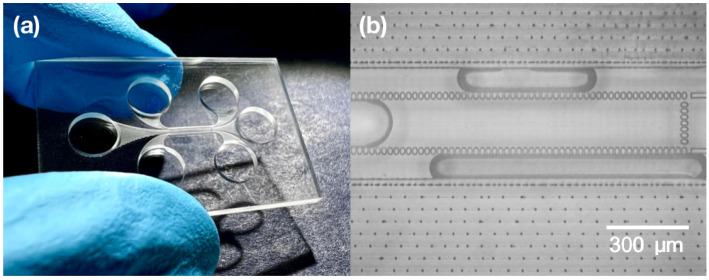
(**a**) Picture of a final liver-on-chip device. (**b**) Optical picture of the produced microfluidic chip part where a channel with a filter and welding seams can be observed.

**Figure 5 materials-16-02174-f005:**
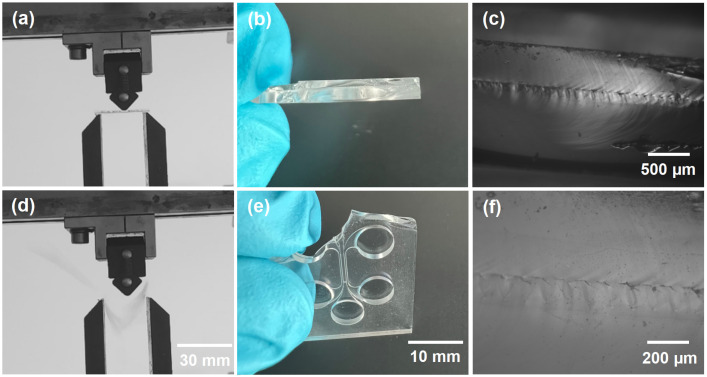
(**a**,**d**) Mechanical resistance test on the chip. (**b**,**e**) Photos of the broken chip after the mechanical resistance test. (**c**,**f**) Optical pictures of the broken structure’s welding seam from the side of the sample.

**Figure 6 materials-16-02174-f006:**
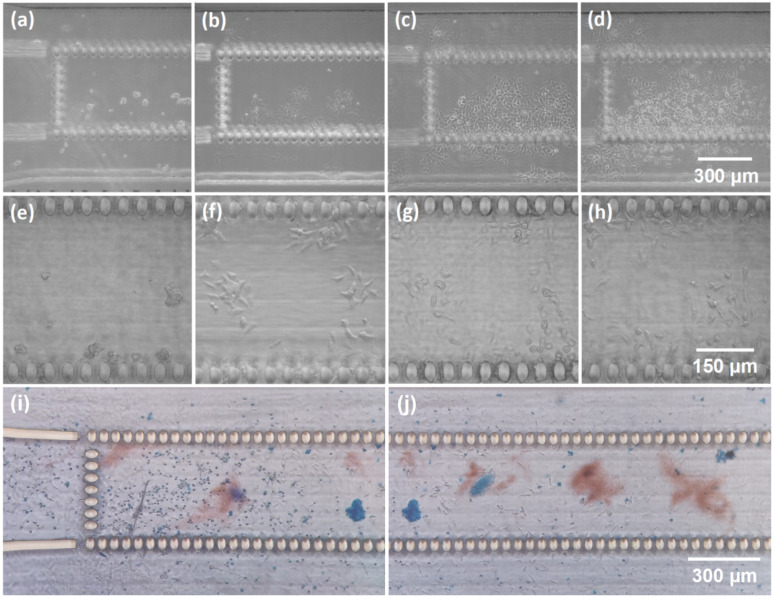
Optical pictures of the fabricated microfluidic channels seeded with HepG2(GS) liver cancer cells (**a**): (**e**) right after the seeding, (**b**,**f**) after 24 h, (**c**,**g**) after 96 h, (**d**,**h**) after 7 days, (**i**,**j**) after 9 days with Trypan Blue.

## Data Availability

Data is contained within the article.

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
