# Peer review of "Combined Femtosecond Laser Glass Microprocessing for Liver-on-Chip Device Fabrication"

_materials, 2023, doi:10.3390/ma16062174_

Round 1

Reviewer 1 Report

Please see the enclosed comment file.

Author Response

File with comments is attached.

Reviewer 2 Report

1.On page 2 line 16, “nanogratings [? ]” should be made certain referenceSame problem is also existing in Line 4 on page 3.

2.On page 2 line 21, “Meanwhile, by forming microvoids, laser ablation [29] or laser welding [30] could be implemented”this sentence should be explained in detail.

3.The surface roughness value of the etched channel should be measured and given out.

4. The seal strength should be measured and given out, for instance the shear test.

5.Why the cell Typan Blue dyeing pictures are not shown listing?

6.Even the references listing should be checked in format and full message.

Author Response

File with comments is attached.
